# A Review of the Role of Wnt in Cancer Immunomodulation

**DOI:** 10.3390/cancers11060771

**Published:** 2019-06-04

**Authors:** Whitney N. Goldsberry, Angelina Londoño, Troy D. Randall, Lyse A. Norian, Rebecca C. Arend

**Affiliations:** 1Department of Obstetrics and Gynecology, University of Alabama at Birmingham, Birmingham, AL 35294, USA; alondono@uabmc.edu (A.L.); rarend@uabmc.edu (R.C.A.); 2Division of Immunology & Rheumatology, University of Alabama at Birmingham, Birmingham, AL 35294, USA; troyrandall@uabmc.edu; 3Department of Nutritional Sciences, University of Alabama at Birmingham, Birmingham, AL 35294, USA; lnorian@uab.edu

**Keywords:** Wnt, β-catenin, immunomodulation, cancer, immunotherapy, tumor microenvironment

## Abstract

Alterations in the Wnt signaling pathway are associated with the advancement of cancers; however, the exact mechanisms responsible remain largely unknown. It has recently been established that heightened intratumoral Wnt signaling correlates with tumor immunomodulation and immune suppression, which likely contribute to the decreased efficacy of multiple cancer therapeutics. Here, we review available literature pertaining to connections between Wnt pathway activation in the tumor microenvironment and local immunomodulation. We focus specifically on preclinical and clinical data supporting the hypothesis that strategies targeting Wnt signaling could act as adjuncts for cancer therapy, either in combination with chemotherapy or immunotherapy, in a variety of tumor types.

## 1. Introduction

The Wnt/β-catenin signaling pathway plays a vital role in many cellular functions. Alterations in the pathway are associated with the advancement of cancers. Wnt pathway aberrations have also been correlated with changes in the tumor microenvironment, such as immune evasion, and this has been the topic of two recent reviews [1,2]. The exact mechanisms by which aberrant Wnt signaling modulates anti-tumor immunity need to be further elucidated to facilitate the optimization of cancer therapies. Here, we provide an overview of Wnt signaling pathways and general anti-tumor immunity, then summarize recent literature illustrating how Wnt modulation of the tumor microenvironment (TME) and immune function are being targeted in an attempt to improve cancer treatment outcomes—including both chemotherapies and immunotherapies—for patients with advanced disease. Although there is evidence linking Wnt signaling with metabolic changes that may also impact anti-tumor immunity, we have not addressed this specific topic here, as these findings were recently discussed in detail elsewhere [3,4].

## 2. The Wnt Pathways

Wnt pathways act as critical signal transduction cascades that modulate embryonic development, adult homeostasis, stem cell control, and wound repair [5]. The first Wnt proteins were discovered in 1982 [6]. There are now 19 known Wnt ligands in mammals, consisting of different glycoproteins, approximately 350–400 amino acids in length, which are highly conserved across many species [7]. The pathways stimulated by these ligands result in gene expression changes that affect the cell’s cytoskeleton and proliferation, while acting as directional growth factors [8]. Due to this important role in cellular regulation, alterations in Wnt signaling consequently may lead to many human diseases, from congenital malformations to nervous system disorders to malignancies, making this pathway a highly desired therapeutic target from a multitude of perspectives. A large portion of Wnt targeted therapies in development focus on Wnt inhibition, while others are Wnt pathway enhancers. A more thorough understanding of the downstream effects of Wnt signaling is necessary for optimal therapeutic manipulation. 

As an initial step in this signaling cascade, Wnt ligands must be processed and exported (Figure 1). Wnt proteins are modified by an attachment of a lipid, palmitoleic acid [9]. This modification is performed by the enzyme Porcupine (PORCN), located in the cellular endoplasmic reticulum [9]. This lipid addition is thought to assist in extracellular membrane attraction and act as a binding motif during ligand–receptor interactions [10]. Once the lipid attachment is completed, Wnt protein is then transported to the plasma membrane for secretion [11]. Specific mechanisms of Wnt extracellular transportation are still being investigated, but these may involve secretory vesicle or exosomal incorporation [12]. 

### 2.1. The Canonical Pathway

Traditionally, Wnt pathways are associated with intracellular β-catenin stabilization [13]. Cytoplasmic accumulation of β-catenin allows for its nuclear translocation, resulting in upregulation of Wnt-responsive genes. The signaling cascade begins when Wnt ligands bind to the transmembrane receptor Frizzled (Fzd or Fz) and the coreceptor low-density lipoprotein receptor-related protein-5/6 (LRP5/6) [14] (Figure 1). After this interaction, the cytoplasmic tail of LRP recruits Axin. Disheveled (Dvl), a protein that may bind to Fzd, may act as a platform for this LRP/Axin interaction [5]. Axin is a component of the β-catenin degradation complex. This complex additionally consists of adenomatous polyposis coli (APC), casein kinase 1α (CK1α), and glycogen synthase kinase 3β (GSK3β). When functioning properly, the β-catenin degradation complex phosphorylates the amino-terminal region of β-catenin. This causes recognition of β-catenin by β-Trcp, an E3 ubiquitin ligase subunit, resulting in the ubiquitination and proteasomal degradation of β-catenin [15,16]. Once the complex is inactive, a stable accumulation of β-catenin occurs in the cell cytosol, culminating in β-catenin translocation into the cell nucleus [17]. Here, β-catenin can bind to the transcription factor T cell factor/lymphoid enhancer-binding factor (TCF/Lef1), activating the transcription of Wnt target genes [18,19]. Axin2 is a generic Wnt transcriptional target gene that is often used as an indicator of activation of this pathway [20].

### 2.2. The Noncanonical Pathways

Most studies have focused on the role of Wnt in the canonical signaling pathway; however, Wnt can also signal through a number of noncanonical pathways. These pathways include the Wnt-JNK, Wnt-RAP1, Wnt-Ror2, Wnt-PKA, Wnt-GSK3MT, Wnt-aPKC, Wnt-RYK, Wnt-mTOR, and Wnt/Ca^2+^ signaling pathways. Most of these pathways overlap to transduce calcium-dependent cell signaling [21]. It is thought that the transmembrane receptor Fz is also involved in these pathways, whereas the coreceptor LRP5/6 is thought to function only in canonical pathway signaling. In fact, there is in vivo evidence to suggest that LRP6 may antagonize noncanonical Wnt pathways, possibly to eliminate competition for Wnt ligands [22]. Ror1/2 is the coreceptor often involved in noncanonical pathways. Ligand interaction with this coreceptor causes an increase in inositol triphosphate (IP3) and diacylglycerol (DAG). IP3 causes a release of Ca^2+^ from the endoplasmic reticulum, leading to activation of protein kinase C (PKC). PKC activates nuclear factor kappa-B (NFκB) and cAMP response element binding protein (CREB). Calcineurin (Cn) and calcium/calmodulin-dependent protein kinase type II (CamKII) are also activated, leading to activation of the nuclear factor of activated T cells (NFAT) and NFκB. NFκB, CREB, and NFAT translocate to the nucleus causing transcription of regulatory genes [21]. Although these many mechanisms may not be fully understood, it is clear that Wnt is involved in vital cellular functions, such as assisting in stabilization of proteins during mitosis, through these alternative pathways [23]. 

### 2.3. Wnt Inhibitors

There are several protein families and genes known to antagonize or modulate Wnt signaling. DKK1, a member of the Dickkopf (DKK) family of proteins, is thought to act as a LRP5/6 ligand antagonist [24] (Figure 1). There are different ideas on the mechanism of this action, with supporting evidence that DKK1 may induce LRP6 internalization and degradation or it may disrupt the Wnt-induced Fz-LRP6 complex [25]. Interestingly, while DKK1 was thought to be tumor suppressive in nature, it is now found to be associated with poor prognosis, supporting tumor growth and metastasis [25]. Contrarily, tumor immune evasion by Dickkopf-related protein 2 (DKK2), with LRP5, is thought to act independently of the Wnt/β-catenin pathway, via inhibition of STAT5 signaling [26]. Sclerostin/SOST proteins are secreted FZD-related proteins. These and Wnt Inhibitory Protein also act as inhibitors by direct interaction with Wnt [8]. Rnf43 and Znrf3 are two Wnt target genes that act as negative-feedback regulators, as they cause degradation of Wnt receptors [27,28,29]. Overall, the roles of these inhibitors and regulators need further investigation, as their ability to modulate abnormal Wnt may be therapeutically useful in a variety of disease settings. 

## 3. Wnt/β-Catenin Signaling and Immunomodulation 

### 3.1. Stem Cells

The Wnt signaling pathway is known to be involved in regulating the self-renewal capacity of non-malignant stem cells. For example, an association between Wnt and adult stem cells was established with the gene disruption of mouse *TCF7L2*. This gene encodes for T cell factor-4 (Tcf-4), which forms the β-catenin/Tcf-4 transcription complex. A study by Korinek et al. demonstrated that cessation of Wnt signaling resulted in a loss of intestinal stem cells, leading to a breakdown of the intestinal epithelium [30]. Wnt signaling has also been found to help maintain the pluripotency of embryonic stem cells [31]. In contrast, DKK1 overexpression, which results in Wnt pathway inhibition, has been shown to eliminate hair follicles and other skin appendages, suggesting a possible blockade of stem cell initiation [32]. Intriguingly, many studies are now analyzing tumor stem cells as a source of immune evasion, as they have been shown to selectively acquire expression of CD80, a surface ligand that dampens immune recognition by binding Cytotoxic T lymphocyte Antigen-4 (CTLA-4) present on activated T cells [33]. Overexpression of WNT5A, a common Wnt ligand, via epigenetic activation in glioblastoma is thought to lead to stem cell differentiation and invasive growth [34]. Additionally, epithelial-to-mesenchymal transition (EMT) and metastasis are supported by WNT5A [35]. In addition, mammary tumor stem cells were found to rely on Wnt proteins as rate-limiting self-renewal signals [36]. There is evidence to support Wnt pathway inhibition with the downregulation of PD-L1 expression, associated with a decreased stemness score signature, in triple negative breast cancer [37]. Potential relationships between cancer stem cells and CD8^+^ tumor infiltration, as related to tumor PD-L1 expression and the effects on cancer progression, are also being investigated [38]. Due to these documented effects of Wnt signaling on promoting stem cell viability and function, it is possible that aberrant Wnt signaling pathways may promote the stem-cell-like qualities of tumor stem cells, thereby facilitating intratumoral immune evasion. 

With its role in cellular regulation, there is ample evidence that Wnt signaling affects hematopoietic stem cells. Wnt stimulation of hematopoietic stem cells may increase their self-renewal capacity [39]. For example, limiting β-catenin activation via noncanonical Wnt signaling stimulation was shown to inhibit the differentiation of hematopoietic stem cells [40,41]. However, other animal studies where β-catenin was mutated did not show a significant change in hematopoiesis [42,43]. Induction of the Wnt/β-catenin pathway through inhibition of GSK3β or the use of WNT3A, a stimulating Wnt ligand, was found to arrest CD8^+^ T cell differentiation into effector cells, promoting self-renewing multipotent CD8^+^ memory stem cells and maintaining the “stemness” of mature memory CD8^+^ T cells [44]. This correlation at the cellular developmental level may provide one of the key links between Wnt signaling and alterations in immune functionality in the tumor. 

### 3.2. Wnt Signaling and Cancer

The associations between Wnt signaling and cancer progression are the topic of intense investigation. This correlation was first established when a WNT factor gene was identified as an oncogene in the mouse mammary cell line RAC311c [45]. Our understanding of these associations has now progressed to include the well-established relationship between APC gene mutations and colorectal cancers. APC gene mutations can be found in most sporadic colorectal cancers, and familial adenomatous polyposis, or the hereditary colon cancer syndrome [46,47]. Alterations in core Wnt regulators were found in a sequencing project of 1134 colorectal cancer samples, noting the incidence of oncogenic Wnt activation in 96% of human colorectal cancers [48]. Axin2 gene mutations have been found in colorectal cancers as well [49]. Hepatocellular carcinomas were found to have Axin1 mutations [50]. Deletions of *GSK3B* may lead hematopoietic stem cells to progress to acute myeloid leukemia [51]. Mutations in these proteins cause a stabilization of cytoplasmic β-catenin due to an inappropriately functioning degradation complex, thus mimicking an upregulation of Wnt signaling. Furthermore, direct mutations in β-catenin have been found in colon cancer and melanoma [52,53]. It is known that β-catenin is also used in cellular adhesion junctions. Through immunohistochemistry evaluation of epithelial ovarian cancer samples, a build-up of β-catenin in the cellular membrane was associated with a decrease in progression-free survival, and resistance to platinum-based chemotherapy [54]. Inactivating gene mutations in the pathway, such as Rnf43 in pancreatic cancer and Znrf3 in adrenocortical cancer or additional cancers, have implicated new links between Wnt signaling and cancer transformation [55,56]. While these direct changes of the Wnt pathway have been established in cancer progression, evolving investigations suggest more depth to Wnt’s role, specifically in regulating anti-tumor immunity. 

### 3.3. Immunity in Cancer

Despite tremendous advancements in traditional chemotherapies, many oncology patients continue to experience rapid progression of disease characterized by chemotherapy resistance, which limits therapeutic options. This has led to the pressing need to investigate alternative methods of treatment, such as immune-directed therapies. Many key aspects of the relationship between the immune system and solid tumors have been elucidated over the last 20 years. There is an appreciable complexity of this system that results in either tumor suppression or tumor progression. It is now established that in order for immune cells to recognize cancer cells and control cancer progression and metastasis, they must first infiltrate the tumor, then remain activated in the TME [57]. The presence and activation of these tumor infiltrating lymphocytes (TILs) typically correlates with tumors that are more sensitive to chemotherapeutic treatment [58,59]. However, if the tumor has been invaded by tumor-promoting T cells, such as Tregs, there may be decreased treatment sensitivity [60]. In particular, higher frequencies of CD8^+^ T cells, and high CD8^+^/CD4^+^ ratios have been correlated with improved overall survival in ovarian cancer [61]. The main goal of immunotherapy is to convert the tumor milieu from an immunologically suppressed state to an inflamed state, for tumor recognition, cell destruction, and improved treatment sensitivity. 

The Cancer Genome Atlas (TCGA) has provided much insight into the role of the immune system in various types of cancers through combined analysis of genomic and patient outcome data. With this resource, TME may be examined from a transcriptional viewpoint, permitting more nuanced cancer categorizations to be made. For example, Thorsson et al. recently performed an immunogenomic analysis of 10,000 tumors from 33 cancer types to identify underlying immune subtypes that are common to all cancers examined [62]. The six subtypes, namely wound healing, interferon-γ dominant, inflammatory, lymphocyte depleted, immunologically quiet and transforming growth factor beta (TGF-β) dominant, help identify the immunological differences present in TME signatures. From this and similar studies, future therapies may be more directed toward the appropriate targets. 

### 3.4. Wnt Signaling and Leukocyte Differentiation

The differentiation of multiple leukocyte populations is regulated by Wnt signaling pathways. Alterations in canonical Wnt signaling may have a genomic influence on T cell development. TCF1 and LEF1 genes have been linked to epigenetic changes that may promote CD8^+^ T cell differentiation by repressing CD4 genetic networks [63,64]. Furthermore, deletions of *Ctnnb1*, the gene encoding for β-catenin, or deletions in *Tcf7*, the gene encoding TCF1, were shown to block thymocyte development [65,66]. When *Tcf7* is deleted in CD8^+^ T cells, functional T cell memory is impaired [67]. However, when p45, a TCF1 variant, is combined with stabilized β-catenin, there is an enhancement of central memory T cell production [68]. Thus, there is support that genomic Wnt pathway alterations correlate with changes in T cell development and differentiation. 

Other immune cell lineages are also influenced by canonical Wnt signaling. Innate lymphoid cells, including natural killer (NK) cells, require TCF1 for development, as shown by the earliest linage-specific precursor expressing high levels of TCF1, and defective NK cell survival in *Tcf7^-/-^* mice [69,70]. In *Lef1^-/-^* and *Fzd9^-/-^* mice, B cell precursors were diminished in the bone marrow [71]. However, B cell development was not impaired despite a *Ctnnb1* deletion in B cell precursors [72]. Furthermore, dendritic cell (DC) differentiation is promoted when there is an upregulation of Fz receptors in DC precursors [73]. The observed results of Wnt signaling alterations throughout multiple cell types remains vast, reflecting the importance of this pathway in immune cellular regulation. 

### 3.5. Wnt Signaling and Immunomodulation

Multiple leukocyte populations within the TME are influenced by both Wnt stimulators and inhibitors. Key Wnt-related immunological alterations are illustrated in Figure 2. For example, the Wnt/β-catenin inhibitor DKK1 has been found to be overexpressed in several different TMEs. High levels of DKK1 were found in serum samples of patients with pancreas, stomach, liver, bile duct, breast, and cervical carcinoma [74]. One might speculate that this could be a negative feedback result of deregulated Wnt signaling from the tumor. However, DKK1 binding to its receptor cytoskeleton associated protein 4 (CKAP4) promoted tumor progression [75]. Additional studies have shown tumor stroma-derived DKK1 targeted β-catenin downregulation in myeloid-derived suppressor cells (MDSCs), leading to an accumulation of these cells, a suppressed T cell response, and tumor proliferation [76]. When a DKK1 vaccination was given in a murine model of myeloma, it was shown to elicit CD4^+^ and CD8^+^ T cell protective immunity [77]. This insinuates a potential for a DKK1 vaccination as an immunotherapeutic adjunct. While the relationship between malignancy and DKK1 remains unclear, there appears to be a strong immunologic influence. 

Noncanonical pathway stimulation may support a tumor proliferative environment. WNT5A stimulation has been shown to increase IL-12 production from DCs, causing an increase in T_H_1 responses [78]. Alternatively, a deficiency of WNT5A in another model showed low levels of interferon gamma producing T_H_1 cells [79]. These Wnt effects on peripheral T cells may alter their functions in tumor recognition. For example, RAR-related orphan receptor C (RARC) was upregulated when CD4^+^ T cells had sustained β-catenin activation, resulting in T_H_17 polarization and production of proinflammatory cytokines that favor tumorigenesis [80]. It has also been shown that Treg survival is increased with increased β-catenin expression [81]. These recognized changes in tumor-infiltrating leukocytes provide insight into the influence of Wnt pathways on tumor immunity and provide a platform for new intervention concepts. 

Immune tolerance is also impacted by Wnt signaling. Irregularities in antitumor cytotoxic T lymphocyte (CTL) priming are associated with Wnt signaling in DCs. The high levels of Wnt ligands found in the TME condition DCs to a regulatory state [82] (Figure 2). This suppressed antitumor immunity was explored via DC-specific LRP5/6 deletions in a murine tumor model. Results showed delayed tumor growth with enhanced effector T cell differentiation, and decreased Treg differentiation [83]. This idea was mimicked pharmacologically via use of the PORCN inhibitor IWP-L6 [83]. In one study, denilukin diftitox (ONTAK; a diphtheria toxin fragment/ IL-2 fusion protein) was given prior to DC vaccinations in patients with melanoma. This resulted in increased β-catenin in the skin and immune tolerance with an increased survival of resting Tregs [84]. In another study, forced expression of non-degradable β-catenin in melanoma cells or DCs led to secretion of the anti-inflammatory cytokine IL-10, which impaired the ability of DCs to cross-prime CD8^+^ CTLs for tumor recognition [85,86]. Through these influences on DCs, related to tumor-induced increases in β-catenin signaling, tumors may acquire tolerogenic characteristics, allowing immune evasion. However, in DC-β-catenin^-/-^ mice, with a CD11c-specific deletion of β-catenin, vaccination with tumor antigen failed to provide tumor protection. Interestingly, these mice were found to be deficient in CD8^+^ T cell immunity [86]. In a mouse model of human melanoma, the β-catenin-dependent transcription blocker, PKF115-584, stimulates DCs to cross-prime tumor-specific CTLs, altering Wnt-induced immunosuppression and improving therapeutic response [85]. These findings suggest complex roles for β-catenin in DCs, where an aberrant amount, through either depletion or overabundance, may lead to immune tolerance. 

Evidence suggests a strong correlation between Wnt pathway changes and immune exclusion. This was recently evaluated with TCGA data. Multiple cancers were analyzed for gene expression of TILs and categorized into a high, intermediate, or low T cell-inflamed tumor environment, based on their expression for genes associated with T cell infiltration. These tumors were then profiled for Wnt/β-catenin-related gene expression profiles. Up to 90% of tumor types showed an inverse correlation between Wnt/β-catenin pathway activation and a T cell-inflamed gene expression signature [87], suggesting that Wnt signaling in the TME suppresses T cell infiltration and/or function. Additional support of immune exclusion was found through a mouse melanoma model that was engineered to express β-catenin. These cells were unable to express C-C motif chemokine ligand 4 (CCL4), leading to decreased CTL infiltration into the TME due to defective recruitment of DCs [88]. Furthermore, in patients with primary and metastatic melanomas treated with BRAF inhibitors, tumor immune infiltration and survival were inversely correlated with β-catenin signaling [89,90]. Collectively, these findings imply a role for altered β-catenin levels in the exclusion of TILs in the TME. 

Immunoevasion mechanisms may also be represented through alterations in immune checkpoint molecules due to Wnt signaling component changes. *GSK3* inactivation in mouse melanoma resulted in a repression of the PD-1 gene, allowing an improved CD8^+^ response [91]. However, in mouse mammary carcinoma, GSK3β was shown to interact with PD-L1, inducing its degradation, which led to increased CTL infiltration [92]. As immune therapies become increasingly important in cancer therapeutic options, these interactions need to be further investigated. 

### 3.6. Immunotherapy

Some immunotherapies are directed at the proteins that regulate T cell function and cytolytic activity. In particular, immune checkpoint inhibitors (ICIs) are monoclonal antibodies against receptors such as CLTA-4 and Programmed Death-1 (PD-1) that act to down-modulate T cell effector function. ICI are now approved for the treatment of malignant melanoma, non-small-cell lung cancer, classical Hodgkin lymphoma, head and neck squamous cell carcinoma, urothelial carcinoma, and renal cell carcinoma [93]. In addition, anti-programmed death-1 ligand (PD-L1) has also shown many promising clinical results [94]. Atezolizumab is an FDA-approved PD-L1 inhibitor that has shown improved progression-free survival and overall survival when used in combination with nabpaclitaxel for metastatic triple negative breast cancer [95]. These results were in patients with known positive PD-L1 tumors. Other anti-PD-L1 therapies include durvalumab and avelumab. Additionally, the monoclonal antibody pembrolizumab (anti-PD-1) was combined with platinum-based chemotherapy in metastatic non-small-cell lung cancer treatment in those who lacked targetable gene mutations. The combination of therapies increased progression-free and overall survival [96]. 

In a similar fashion, ipilimumab and tremelimumab are monoclonal antibodies to CTLA4, which normally functions to restrain effector T cell activation. Ipilimumab is known for its significant progression-free and overall survival advancement in melanoma [97]. Further improvements in metastatic melanoma were seen following a combination of ipilimumab and anti-PD-L1 therapy [98]. 

With the rising use of mono- and combinatorial immunotherapy in the clinical setting, there is a need to further understand the TME in patients who do not respond. It has been reported that decreased TILs result in resistance to ICIs [99]. Wnt pathway regulation may play a role in this lack of response, as tumor-intrinsic, active β-catenin signaling in human melanoma has been associated with resistance to anti-CTLA4 and anti-PD-L1 antibodies due to T cell exclusion [88]. It has also been speculated that tumor cells may lack the antigens needed for recognition by TILs, perhaps leading to immunotherapy resistance in these cancers. This was investigated with an analysis of 266 melanoma tumor samples from TCGA. Tissues were divided into categories based on a high or low expression of genes that were associated with T cell infiltration. This was correlated with nonsynonymous somatic mutations as a representative of mutational neoantigens that the tumor may possess. It was concluded that the change in tumor gene expression of infiltrating T cells did not correlate with increased mutational neoantigens [100]. However, gene signatures of these melanoma tissues did support prior evidence of a correlation between Wnt/β-catenin pathway activation and a reduction in T cell infiltration in the tumor [100]. This evidence, in addition to prior stated support, shows a strong correlation between increased Wnt signaling and decreased T cell infiltration in tumors. 

## 4. Targeting Wnt Signaling as a Novel Therapeutic Option

There are numerous relationships between Wnt signaling, immune function, and cancer progression; most notable is that of increased Wnt signaling correlating with decreased tumor T cell infiltration, as discussed above. Wnt pathways have a vast number of roles that offer multiple options for pathway modulation in the malignant setting. The United States Patent and Trade Office Patent and Patent Application databases report 103 unique Wnt signaling modulators being investigated, with 34 in clinic trials [101]. Actions of these therapies range from the signal pathway component targeting activators to Wnt inhibitors. Many intriguing investigations are evaluating tumor immunity with drug intervention. Understanding the immunomodulation of these therapies will be essential during attempts to transition the TME to a less resistant milieu. 

Numerous studies are currently combining Wnt therapies with ICIs, additional Wnt inhibitors, or chemotherapies in an attempt to achieve optimal effects on tumor control. However, it is currently unknown how the combination of therapies will affect tumor progression. In colon cancer, changes in Wnt signaling pathways have shown counter-intuitive results in some studies, such as WNT-TCF blockade actually boosting metastasis [102]. In contrast, other results have suggested a synergy between Wnt pathway inhibition and ICIs, for example when used as a combinatorial therapy in a mouse melanoma model [103]. Additional combination treatments with multiple Wnt inhibitors have been found to revert resistance and repress tumor growth in colorectal cancer [104]. Furthermore, the combination of Wnt antagonists with taxane therapies elicited a synergistic effect by sensitizing cancer stem cells to taxane-induced death [105]. Table 1 identifies clinical trials of Wnt modulators, with the listed agent, mechanism of action, intervention strategy, and targeted disease. 

Therapeutic modulation of Wnt signaling is being investigated through many avenues, several of which are depicted in Figure 1. One agent under investigation is DKN-01. This is a monoclonal antibody to DKK1, the Wnt/β-catenin inhibitor. DKN-01 is being used in several clinical trials for investigation of safety and efficacy in patients with multiple primary tumor types. There have been two completed clinical trials with this drug in multiple myeloma (NCT01711671, NCT01457417). Results are available from one of these studies, but published conclusions are pending [106]. High serum levels of DKK1 were found in patients with pancreas, stomach, liver, bile duct, breast, and cervical cancers [74]. Prior studies have also shown that increased DKK1 stabilized MDSC populations, leading to suppression of the T cell intratumoral response [76]. The direct influence on Wnt signaling from these agents is convoluted given it is a result of inhibition of a Wnt inhibitor. If this inhibitor is blocking canonical Wnt signaling, it may lead to an upregulation of noncanonical signaling. Perhaps with inhibition of DKK1, the alternative pathway will be normalized. There may also be a decrease in intratumoral suppressive MDSCs, resulting in increased tumor cell recognition and clearance by CD8^+^ T cells. It will be interesting to view future clinical trial results related to effects on the TME that occur following targeted inhibition of a Wnt inhibitor, with and without combination therapy. 

Alternative therapeutic agents act directly on Wnt ligand secretion. PORCN inhibitors are known to block the extracellular excretion of Wnt by blocking the enzyme responsible for palmitoylation of Wnt ligands (Figure 1). This family of inhibitors includes C59, CGX 1321, ETC1922159, LGK974, IWP-L6, and RXC004. With this overall extracellular decrease in Wnt ligands, the TME may have the ability to convert to a T cell-inflamed environment, based on evidence of increased Wnt signaling correlated with T cell-noninflamed tumors [87]. Many of these molecules remain under investigation in the preclinical setting; however, some studies have advanced to clinical trials. Pending results will provide insight into treatment efficacy for multiple malignancies. 

Some Wnt altering agents have been previously approved in nonmalignant diseases. Artesunate is a compound extracted from the herb *Artemisia annua*, used as an FDA-approved antimalarial drug. Treatment of colorectal tumor xenografts with this agent correlated with decreased growth of tumors with inhibition of a hyperactive Wnt/β-catenin pathway [107]. The exact mechanism of the agent remains unknown. However, two phase 1 trials are now completed using this agent in subjects with hepatocellular carcinoma or solid tumors. Results are pending from these dose-escalation studies (NCT02304289, NCT02353026). An additional completed phase 1 study evaluated artesunate as an add-on therapy in subjects with metastatic or locally advanced breast cancer, with unreleased results (NCT00764036). Additionally, niclosamide is an anti-helminthic agent that has been identified to have many molecular targets, including inhibition of the Wnt pathway. Specifically, the Axin-GSK3β interaction is targeted in this pathway, resulting in a suppression of Wnt/Snail [108]. There are several phase 1 trials involving this therapeutic agent in various cancers. One trial involving several types of prostate cancer has been completed, with pending results (NCT02532114). One note of caution is that with so many known targets, it may be hard to determine if the effects of these therapies are directly related to Wnt changes in the tumor, as opposed to additional mechanistic alterations.

Additional Wnt-inhibiting agents are being tested in clinical trials. Ipafricept, also known as OMP54F28, is a recombinant fusion protein with an extracellular Fzd 8 receptor portion attached to an IgG1 Fc fragment, which acts as a decoy receptor for Wnt ligands [109]. Four phase 1 trials have been completed with this therapy (NCT02069145, NCT02092363, NCT02050178, NCT01608867). Study conclusions have not been released. WNT5A is a Wnt ligand mimicked by Foxy-5, a formylated 6 amino acid peptide fragment. The agent is thought to impair migration of epithelial cancer cells, giving it anti-metastatic potential [110]. Two clinical trials have been completed to determine appropriate doses for phase 2 trials (NCT02020291, NCT02655952). Due to increased β-catenin levels found in many colon cancers, a CREB-binding protein (CBP)/catenin inhibitor, PRI724, is being investigated [111]. Two phase 1 clinical trials have been completed with the use of this inhibitor in pancreatic cancers and acute and chronic myeloid leukemias (NCT01764477, NCT01606579). Results from these clinical trials are currently unavailable. SM08502 is an orally bioavailable small molecule inhibitor that is thought to inhibit the expression of Wnt signaling pathway genes, but further investigation is being elucidated on the exact mechanisms of action and its relation to Wnt. One phase 1 clinical trial is using this agent in solid tumors (NCT03355066). Completion of these studies, and future studies, may provide insight into the optimal dose and timing for administration of Wnt-based therapeutics and malignancies that are most sensitive to these agents. 

## 5. Conclusions

Here, we have reviewed preclinical and clinical data that support the continued investigation of combining Wnt signaling modulators with cancer chemotherapies or immunotherapies, in order to achieve better tumor control in a greater percentage of patients. As discussed, Wnt signaling is involved in regulating a wide variety of complex cellular functions, in both malignant cells and leukocytes. Furthermore, aberrations in Wnt signaling are now well-established in a multitude of malignancies and elevated Wnt signaling shows a strong correlation with overall immune suppression. There is also evidence to suggest that tumor stem cell promotion, via enhanced Wnt signaling, may contribute to immune evasion. Thus, finding ways to alter Wnt signaling specifically in tumor cells and/or tumor stem cells could provide a platform for inducing beneficial changes in the immune-related TME, leading to improved cancer treatment efficacy. Wnt-modulating agents on clinical trial include DKK1 antibodies, PORCN inhibitors, AXIN1 activators, Wnt decoys, WNT5A mimics, β-catenin inhibitors, among others with less clearly-defined modes of action. At this time, additional studies are needed to more fully understand how Wnt-modulating agents are altering the intratumoral immune response and broader TME. However, the synergistic effects seen to date with ICIs or chemotherapies used in combination with Wnt inhibitors lends validity to the idea that targeting the Wnt pathway is a promising therapeutic approach for many tumor types, as doing so may promote protective anti-tumor immunity and convert the tumor milieu to one more susceptible to traditional therapies. 

## Figures and Tables

**Figure 1 cancers-11-00771-f001:**
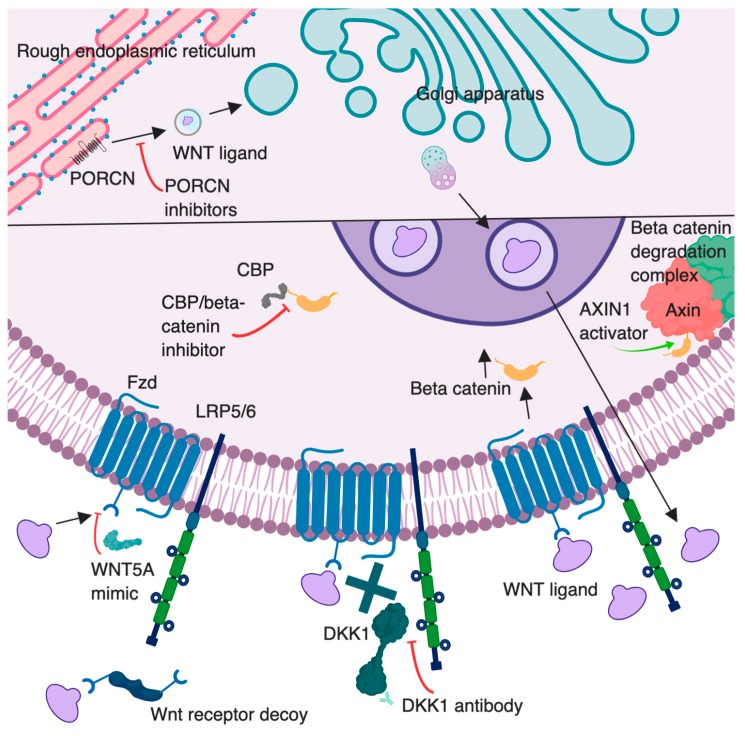
Overview of the canonical Wnt signaling pathway illustrating therapeutic intervention points targeted by Wnt modulators. The lower half of the figure represents a zoomed in portion of the cell. Porcupine (PORCN) inhibitors (e.g., LGK974) block secretion of WNTs by inhibiting their palmitoylation. AXIN1 activators (e.g., niclosamide) promote β-catenin degradation. Dickkopf 1 (DKK1) antibodies may increase Wnt/β-catenin signaling by blocking DKK1 binding to LRP5/6; beneficial effects in cancer may be through inhibition of DKK1 binding to CKAP4 or indirect effects on immune cells. Wnt receptor decoys (e.g., OMP54F28) prevent Wnt binding to Fzd receptors. WNT5A mimic (Foxy-5) is a peptide that activates noncanonical Wnt signals. CBP/beta-catenin inhibitors (e.g., PRI-724) disrupt the interaction between CREB-binding protein (CBP) and β-catenin. Created with BioRender.com. Proteins and cell compartments are not drawn to scale.

**Figure 2 cancers-11-00771-f002:**
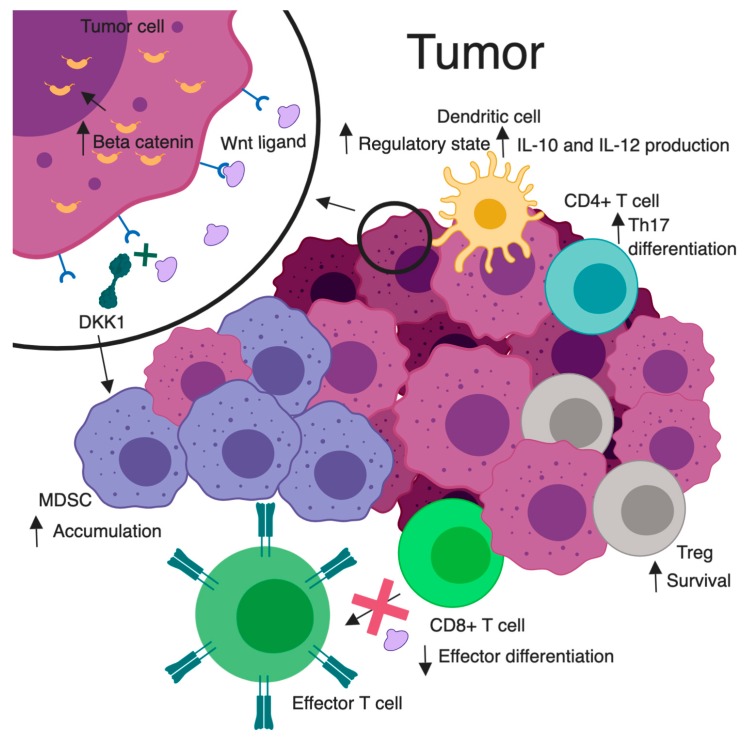
Wnt/β-catenin signaling and tumor immunomodulation. A magnified tumor cell (top left), illustrates Wnt signaling leading to elevated cytosolic and nuclear β-catenin. Increased Wnt signaling is associated with heightened survival of Tregs, skewed differentiation of CD4^+^ T cells to a pro-tumorigenic Th17 subtype, conversion of dendritic cells to a regulatory state with enhanced IL-10 and IL-12 secretion, and decreased effector differentiation and function in CD8^+^ T cells. DKK1 inhibits Wnt ligand/receptor interactions. Elevated DKK1 leads to an accumulation of MDSC in the TME and subsequent inhibition of effector CD8^+^ T cell function. Created with BioRender.com. Not drawn to scale.

**Table 1 cancers-11-00771-t001:** Cancer Clinical Trials of Wnt Modulators. Agents used are listed with the mechanism of action, the corresponding investigated disease, the phase of the trial, the investigational intervention, and the clinical trial identifier [106].

Agent	Mechanism of Action	Disease	Phase	Therapy	Identifier
DKN-01	DKK1 antibody	Esophageal Neoplasms	1	Dose escalation in combination with paclitaxel or pembrolizumab	NCT02013154
		Adenocarcinoma of the Gastroesophageal Junction			
		Gastroesophageal Cancer			
		Squamous Cell Carcinoma			
		Gastric Adenocarcinoma			
DKN-01	DKK1 antibody	Endometrial Cancer	2	Monotherapy or in combination with paclitaxel	NCT03395080
		Uterine Cancer			
		Ovarian Cancer			
DKN-01	DKK1 antibody	Hepatocellular Carcinoma	1, 2	Phase 1/2 as a monotherapy or combination with sorafenib	NCT03645980
DKN-01	DKK1 antibody	Multiple Myeloma	1	Pilot study of combination with lenalidomide/dexamethasone	NCT01711671
DKN-01	DKK1 antibody	Multiple Myeloma	1	Dose escalation	NCT01457417
		Solid Tumors			
		Non-Small Cell Lung Cancer			
DKN-01	DKK1 antibody	Carcinoma of Intrahepatic and Extra-hepatic Biliary System	1	Dose escalation combined with gemcitabine and cisplatin	NCT02375880
		Carcinoma of Gallbladder			
		Bile Duct Cancer			
		Cholangiocarcinoma			
CGX 1321	PORCN inhibitor	Colorectal Adenocarcinoma	1	Single agent dose escalation	NCT03507998
		Gastric Adenocarcinoma			
		Pancreatic Adenocarcinoma			
		Bile Duct Carcinoma			
		Hepatocellular Carcinoma			
		Esophageal Carcinoma			
		Gastrointestinal Cancer			
CGX 1321	PORCN inhibitor	Solid Tumors	1	Single agent dose escalation with or without pembrolizumab	NCT02675946
		GI Cancer			
ETC1922159	PORCN inhibitor	Solid Tumors	1	Single agent dose escalation	NCT02521844
LGK974	PORCN inhibitor	Pancreatic Cancer	1	Single agent and in combination with PDR001	NCT01351103
		BRAF Mutant Colorectal Cancer			
		Melanoma			
		Triple Negative Breast Cancer			
		Head and Neck Squamous Cell Cancer			
		Cervical Squamous Cell Cancer			
		Esophageal Squamous Cell Cancer			
		Lung Squamous Cell Cancer			
RXC004	PORCN inhibitor	Cancer	1	Dose tolerability	NCT03447470
		Solid Tumor			
Artesunate	Unknown	Hepatocellular Carcinoma	1	Single agent dose escalation	NCT02304289
Artesunate	Unknown	Colorectal Cancer	2	Neoadjuvant single agent	NCT03093129
Artesunate	Unknown	Solid Tumors	1	Single agent dose escalation	NCT02353026
Artesunate	Unknown	Colorectal Cancer	2	Neoadjuvant single agent	NCT02633098
		Bowel Cancer			
Artesunate	Unknown	Metastatic Breast Cancer	1	Add-on therapy	NCT00764036
		Locally Advanced Breast Cancer			
Niclosamide	AXIN1 activator	Colon Cancer	1	Dose escalation	NCT02687009
Niclosamide	AXIN1 activator	Metastatic Prostate Carcinoma	1	Dose escalation with enzalutamide	NCT03123978
		Recurrent Prostate Carcinoma			
		Stage IV Prostate Cancer			
Niclosamide	AXIN1 activator	Castration-Resistant Prostate Carcinoma	1	Dose escalation with enzalutamide	NCT02532114
		Metastatic Prostate Carcinoma			
		Recurrent Prostate Carcinoma			
		Stage IV Prostate Adenocarcinoma			
Niclosamide	AXIN1 activator	Colorectal Cancer	2	Single agent	NCT02519582
Niclosamide	AXIN1 activator	Metastatic Prostate Cancer	2	Combination with abirateronae acetate and prednisone	NCT02807805
		Recurrent Prostate Cancer			
		Stage IV Prostate Cancer			
OMP54F28	Wnt receptor decoy	Hepatocellular Cancer	1	Dose escalation with sorafenib	NCT02069145
		Liver Cancer			
OMP54F28	Wnt receptor decoy	Ovarian Cancer	1	Combined with paclitaxel and carboplatin	NCT02092363
OMP54F28	Wnt receptor decoy	Pancreatic Cancer	1	Combined with Nab-paclitaxel and gemcitabine	NCT02050178
		Stage IV Pancreatic Cancer			
OMP54F28	Wnt receptor decoy	Solid Tumors	1	Dose escalation	NCT01608867
Foxy-5	WNT5A mimic	Metastatic Breast Cancer	1	Dose escalation	NCT02020291
		Colorectal Cancer			
		Prostate Cancer			
Foxy-5	WNT5A mimic	Metastatic Breast Cancer	1	Dose escalation	NCT02655952
		Metastatic Colon Cancer			
		Metastatic Prostate Cancer			
PRI724	CBP/catenin inhibitor	Advanced Pancreatic Cancer	1	Dose escalation with gemcitabine	NCT01764477
		Metastatic Pancreatic Cancer			
		Pancreatic Adenocarcinoma			
PRI724	CBP/catenin inhibitor	Acute Myeloid Leukemia	1, 2	Dose escalation, combined with dasatinib for CML or cytarabine for AML	NCT01606579
		Chronic Myeloid Leukemia			
SM08502	Unknown	Solid Tumors, Adult	1	Single agent dose escalation	NCT03355066

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
