# Peer review of "A Review of the Role of Wnt in Cancer Immunomodulation"

_cancers, 2019, doi:10.3390/cancers11060771_

Round 1

Reviewer 1 Report

This manuscript reviews the role of (intra)tumoral Wnt signaling in immune modulation and suppression, and makes the case for a therapeutic strategy that targets Wnt signaling in combination with other (immune-) therapies.

The subject is broad, touching on several fields (Wnt signaling, oncology, immunology and cancer immunity), which the authors necessarily introduce in very broad strokes.

In general it is well-written and--I think--a good addition to the literature, even if it is only 2 years after a similar review (Gopalkrisha 2017. J Hematol-Oncol 10:101).

However, I see two missed opportunities:

1) The figure is not of good quality. It also does not add value. It would be better to make 1 or 2 (additional) images on imunosuppressive mechanisms, and/or therapeutic strategies. (This was done much better in the 2017 review.)

2) The ms is informative, yes, and able to raise some excitement about therapeutic potential. Yet it does not go below the surface; I miss a conceptual synthesis. Indeed, the conclusion reads as a summary of a summary—confirming a state of complexity and confusion, rather than an attempt to clarify the field somewhat, or an effort to distill insight. Perhaps the authors can strengthen this section.

A minor thing: regarding Ref [11], this paper does not show release of b-cat by Dishevelled from the destruction complex. Please rephrase or add the proper reference (Please double check citation accuracy elsewhere).

Author Response

This manuscript reviews the role of (intra)tumoral Wnt signaling in immune modulation and suppression, and makes the case for a therapeutic strategy that targets Wnt signaling in combination with other (immune-) therapies.

The subject is broad, touching on several fields (Wnt signaling, oncology, immunology and cancer immunity), which the authors necessarily introduce in very broad strokes.

In general it is well-written and--I think--a good addition to the literature, even if it is only 2 years after a similar review (Gopalkrisha 2017. J Hematol-Oncol 10:101).

Author response: Thank you for your generous comments. 

However, I see two missed opportunities:

1) The figure is not of good quality. It also does not add value. It would be better to make 1 or 2 (additional) images on imunosuppressive mechanisms, and/or therapeutic strategies. (This was done much better in the 2017 review.)

Author response: We agree that the original figure was not of sufficient quality.  Therefore, we have completely re-made Figure 1 with improved graphics.  Additionally, we have added a new Figure 2 that summarizes Wnt signaling and immunomodulation.

2) The ms is informative, yes, and able to raise some excitement about therapeutic potential. Yet it does not go below the surface; I miss a conceptual synthesis. Indeed, the conclusion reads as a summary of a summary—confirming a state of complexity and confusion, rather than an attempt to clarify the field somewhat, or an effort to distill insight. Perhaps the authors can strengthen this section.

Author response: We have improved the clarity of the Conclusion section to provide a more cohesive summary of our review and the topic as a whole.

A minor thing: regarding Ref [11], this paper does not show release of b-cat by Dishevelled from the destruction complex. Please rephrase or add the proper reference (Please double check citation accuracy elsewhere).

Author response: Thank you for noticing the citation error. This has been corrected and the remainder of the references verified.

Reviewer 2 Report

Summary: The authors have written a quality review of known and possible roles of Wnt signaling in cancer immunomodulation.  They begin with a brief but sound review of the basics of Wnt signaling, with an emphasis on canonical Wnt signaling but also describing non-canonical, and they then describe how Wnt signaling affects immunomodulation and immunotherapy in cancer, and end with comments about targeting Wnt signaling in cancer.

Broad suggestions:  I have three suggestions for improvement.  First, an additional figure that summarizes possible Wnt signaling-immune system interactions would be helpful.  Second, in the section on targeting Wnt for therapy, it would be helpful to discuss more how these therapies could in theory more specifically affect the immune response.  Third is the issue of side effects of using Wnt modulation to affect the immune system (related to point two).  The authors do note that given the wide range of Wnt effects it may be difficult to separate direct effects on the tumor to other mechanisms.  It is possible that Wnt-mediated tumor-targeting and immune targeting could be synergistic, but it is also possible they can conflict - what helps with one may negatively affect the other.  Note that Wnt inhibition is not always the best option for colorectal cancer:

https://journals.plos.org/plosone/article?id=10.1371/journal.pone.0150697

Specific: Point  three may be best inserted after lines 309-311 in the text.

Author Response

Summary: The authors have written a quality review of known and possible roles of Wnt signaling in cancer immunomodulation.  They begin with a brief but sound review of the basics of Wnt signaling, with an emphasis on canonical Wnt signaling but also describing non-canonical, and they then describe how Wnt signaling affects immunomodulation and immunotherapy in cancer, and end with comments about targeting Wnt signaling in cancer.

Author response: Thank you for your generous comments.

Broad suggestions:  I have three suggestions for improvement.  First, an additional figure that summarizes possible Wnt signaling-immune system interactions would be helpful.  Second, in the section on targeting Wnt for therapy, it would be helpful to discuss more how these therapies could in theory more specifically affect the immune response.  Third is the issue of side effects of using Wnt modulation to affect the immune system (related to point two).  The authors do note that given the wide range of Wnt effects it may be difficult to separate direct effects on the tumor to other mechanisms.  It is possible that Wnt-mediated tumor-targeting and immune targeting could be synergistic, but it is also possible they can conflict - what helps with one may negatively affect the other.  Note that Wnt inhibition is not always the best option for colorectal cancer:

https://journals.plos.org/plosone/article?id=10.1371/journal.pone.0150697

Specific: Point  three may be best inserted after lines 309-311 in the text.

Author response:

First suggestion: We agreed that an additional figure summarizing Wnt signaling and immunomodulation was helpful. This has been added as Figure 2. Additionally, Figure 1 has been redone for improved design.

Second suggestion: In the targeting Wnt for therapy section, immune responses were added to the discussion.

Third suggestion: Thank you for the reference. Point three was added where suggested, addressing that synergy may not be the outcome of combined treatments. This was elaborated upon further in the text.

Reviewer 3 Report

This review discusses a very timely subject, namely ICI non-responsiveness and the role of the Wnt signaling cascade. This has been the subject of several recent reviews, including: Wang B, Tian T, Kalland KH, Ke X, Qu Y. Targeting wnt/beta-catenin signaling for cancer immunotherapy. Trends in pharmacological sciences. 2018;39:648-658 and Trends Cell Biol. 2019 Jan;29(1):44-65. doi: 10.1016/j.tcb.2018.08.005. Epub  2018 Sep 13. WNT Signaling in Cancer Immunosurveillance. Galluzzi L1, Spranger S2, Fuchs E3, López-Soto A4, which should be cited. The table of various inhibitors should be complemented with a diagram/figure showing where (if known the various inhibitors work, i.e. extracellular, cytoplasm or nuclear targets). A minor point, but PRI-724 is techinically a CBP/catenin inhibitor (i.e binds to CBP) and not a beta-catenin inhibitor per se.

A discussion of two other aspects, Wnt signaling and metabolism, and how that might effect the immune response in the tumor microenvironment would be beneficial as well as a discussion of cancer stem cells and Wnt and their potential role in immune evasion especially given several recent papers in this regard i.e.

Adaptive Immune Resistance Emerges from Tumor-Initiating Stem Cells. Miao Y, Yang H, Levorse J, Yuan S, Polak L, Sribour M, Singh B, Rosenblum M, Fuchs E. Cell. 2019 Apr 22. pii: S0092-8674(19)30287-9. doi: 10.1016/j.cell.2019.03.025 and 16.

WNT signaling modulates PD-L1 expression in the stem cell compartment of triple-negative breast cancer.Castagnoli L, Cancila V, Cordoba-Romero SL, Faraci S, Talarico G, Belmonte B, Iorio MV, Milani M, Volpari T, Chiodoni C, Hidalgo-Miranda A, Tagliabue E, Tripodo C, Sangaletti S, Di Nicola M, Pupa SM. Oncogene. 2019 Jan 31. doi: 10.1038/s41388-019-0700-2 and Low CD8⁺ T Cell Infiltration and High PD-L1 Expression Are Associated with Level of CD44⁺/CD133⁺ Cancer Stem Cells and Predict an Unfavorable Prognosis in Pancreatic Cancer.Hou YC, Chao YJ, Hsieh MH, Tung HL, Wang HC, Shan YS.Cancers (Basel). 2019 Apr 15;11(4). pii: E541.

Finally, although perhaps not simple to address as Wnt signaling is enormously complex, a comment regarding how both inhibitors of Wnt signaling, as well as in principle activators of Wnt signaling (i.e. Abs of the Wnt inhibitor DKK1) could potentially both have beneficial effects would be useful for readers.

Author Response

This review discusses a very timely subject, namely ICI non-responsiveness and the role of the Wnt signaling cascade. This has been the subject of several recent reviews, including: Wang B, Tian T, Kalland KH, Ke X, Qu Y. Targeting wnt/beta-catenin signaling for cancer immunotherapy. Trends in pharmacological sciences. 2018;39:648-658 andTrends Cell Biol. 2019 Jan;29(1):44-65. doi: 10.1016/j.tcb.2018.08.005. Epub  2018 Sep 13. WNT Signaling in Cancer Immunosurveillance. Galluzzi L1, Spranger S2, Fuchs E3, López-Soto A4, which should be cited. The table of various inhibitors should be complemented with a diagram/figure showing where (if known the various inhibitors work, i.e. extracellular, cytoplasm or nuclear targets). A minor point, but PRI-724 is techinically a CBP/catenin inhibitor (i.e binds to CBP) and not a beta-catenin inhibitor per se.

Author response: Thank you for your generous comments.  Figure 1 was redone for improved visualization.  We have added a new Figure 2 that summarizes Wnt signaling and immunomodulation.  We think with this additional figure, readers will have more clarification of inhibitors. DKK1 was added in Figure 2. Due to this, an additional figure only addressing inhibitors was not added.  Thank you for recognizing our mistake of PRI-724. This was corrected.

A discussion of two other aspects, Wnt signaling and metabolism, and how that might effect the immune response in the tumor microenvironment would be beneficial as well as a discussion of cancer stem cells and Wnt and their potential role in immune evasion especially given several recent papers in this regard i.e.

Adaptive Immune Resistance Emerges from Tumor-Initiating Stem Cells. Miao Y, Yang H, Levorse J, Yuan S, Polak L, Sribour M, Singh B, Rosenblum M, Fuchs E. Cell. 2019 Apr 22. pii: S0092-8674(19)30287-9. doi: 10.1016/j.cell.2019.03.025 and 16.

WNT signaling modulates PD-L1 expression in the stem cell compartment of triple-negative breast cancer.Castagnoli L, Cancila V, Cordoba-Romero SL, Faraci S, Talarico G, Belmonte B, Iorio MV, Milani M, Volpari T, Chiodoni C, Hidalgo-Miranda A, Tagliabue E, Tripodo C, Sangaletti S, Di Nicola M, Pupa SM. Oncogene. 2019 Jan 31. doi: 10.1038/s41388-019-0700-2 and Low CD8Cell Infiltration and High PD-L1 Expression Are Associated with Level of CD44/CD133 Cancer Stem Cells and Predict an Unfavorable Prognosis in Pancreatic Cancer.Hou YC, Chao YJ, Hsieh MH, Tung HL, Wang HC, Shan YS.Cancers (Basel). 2019 Apr 15;11(4). pii: E541.

Author response: The stem cell references were very helpful.  A section on stem cells and Wnt correlation has been added, relating this to immune evasion.  We appreciate the suggestion of adding a section on Wnt signaling and metabolism; however, instead of trying to summarize this important topic in one or two paragraphs, we instead mentioned that this topic has been expertly and recently reviewed.

Finally, although perhaps not simple to address as Wnt signaling is enormously complex, a comment regarding how both inhibitors of Wnt signaling, as well as in principle activators of Wnt signaling (i.e. Abs of the Wnt inhibitor DKK1) could potentially both have beneficial effects would be useful for readers. 

Author response: Additional comments were added for clarification on activators and inhibitors of Wnt.